# The Effect of Sandblasting on Properties and Structures of the DC03/1.0347, DC04/1.0338, DC05/1.0312, and DD14/1.0389 Steels for Deep Drawing

**DOI:** 10.3390/ma14133540

**Published:** 2021-06-25

**Authors:** Janusz Krawczyk, Michał Bembenek, Łukasz Frocisz, Tomasz Śleboda, Marek Paćko

**Affiliations:** 1Faculty of Metals Engineering and Industrial Computer Science, AGH University of Science and Technology, A. Mickiewicza 30, 30-059 Kraków, Poland; jkrawcz@agh.edu.pl (J.K.); lfrocisz@agh.edu.pl (Ł.F.); sleboda@agh.edu.pl (T.Ś.); packo@metal.agh.edu.pl (M.P.); 2Faculty of Mechanical Engineering and Robotics, AGH University of Science and Technology, A. Mickiewicza 30, 30-059 Kraków, Poland

**Keywords:** sandblasting, DC03/1.0347, DC04/1.0338, DC05/1.0312, DD14/1.0389, microstructure, deep drawing steel

## Abstract

The erosion phenomenon has a significant influence on many metallic materials used in numerous industrial sectors. In this paper, we present the results of an analysis of the influence of abrasive impact erosion on surface and properties of DC03/1.0347, DC04/1.0338, DC05/1.0312, and DD14/1.0389 deep drawing steels. The chemical composition, static tensile tests, hardness tests, drawability tests, erosion tests, microstructure analysis, surface roughness, and hardness of the plates were investigated. The wear mechanisms and wear behavior of the investigated steels were also discussed. The results obtained in this study allowed the assessment of the microstructural changes in deep drawing steels under the influence of intense erosive impact. The obtained results indicate that the erosive impact may cause a significant grain refinement of the microstructure of the surfaces of the investigated materials. Moreover, large amounts of heat released during erosive impact may cause the material phase changes. This research expands the knowledge on specific mechanisms taking place during sandblasting and their influence on the properties of deep drawing steels and their wear behavior.

## 1. Introduction

The erosion phenomenon has a significant influence on many metallic materials used in many industrial sectors [1,2]. Sandblasting is the operation of applying a stream of abrasive material under high pressure against a surface to clean it or change its geometry or properties. It is widely used prior to the application of various coatings to the surface of a material. The most commonly used abrasive material is sand; however, other materials can also be used, such as glass or metals. Sandblasting the surfaces of various structural materials is often recommended prior to adhesive bonding, these materials include steel [3], titanium or titanium alloys [4], aluminum alloys [5], and composite materials [6]. The effects of various sandblasting parameters such as pressure, angle, and sand granulometry on galvanized steel surface chemical composition and surface roughness for the purpose of increasing its adhesion properties for further use in structural bonding applications were investigated in [7]. The influence of time of sandblasting, as a surface mechanical attrition treatment method, on the corrosion behavior of carbon steel in different solutions with different pHs was studied in [8]. The microstructure, surface morphologies, and corrosion behavior of 316L stainless steel processed by sandblasting were investigated in [9]. Some of the investigations have pointed out the significance of erodent particle characteristics and the effect of particle size and shape on the erosion of 18Ni(250) maraging steel [10]. The effects of sand impact angle and velocity on the erosion rate of the investigated S45C steel has been presented in [11]. The results of the investigations concerning the influence of sandblasting on tensile and fatigue properties of API 5L X52 were presented in [12]. Sandblasting may strongly influence the microstructure of the surface of the materials subjected to such treatment [13,14], leading, in some cases, to its nanocrystallization [15,16,17]. The formation of ultrafine or nanocrystalline microstructure in the surface layers of the materials often leads to considerable improvements in their tribological and mechanical properties [18,19]. Strain-induced grain refinement resulting from sandblasting has been reported in many studies [20,21,22]. The effect of sandblasting on surface properties of C45 steel sheet for adhesion was investigated in [23]. The investigated parameters included pressure and four types of abrasive materials, each material type being of a different granulation. Many studies have been conducted to understand the influence of sandblasting parameters on the erosion process or surface integrity of a material after sandblasting [24,25], but the information concerning the influence of material surface behavior resulting from sandblasting on its properties and wear behavior [26,27,28] is still limited. This study aimed at determining the resistance of selected deep-drawing steel sheets to erosive wear; DC03, DC04, DC05, and DD14 steel sheets were investigated. The resistance to erosive wear referred to selected mechanical properties, microstructure, and chemical composition of the sheets. The wear mechanisms and wear behavior of the investigated steels were also discussed.

## 2. Materials and Methods

For the investigations, 1.5 mm (±0.05 mm) thick sheets from four grades of deep drawing steels were used. The samples were made of DC03/1.0347, DC04/1.0338, DC05/1.0312 steel produced by cold rolling, and DD14/1.0389 steel produced by hot rolling. The steels used in the experiments were in an aged condition (cut into sheets and stored in such a state at room temperature until the tests were performed), that is, more than 6 months elapsed from the date of production to the performance of the tests. The material in this condition was selected for the purpose of determining the possibility of the regeneration (repair) process of structural elements made of such steels.

### 2.1. Chemical Composition Analysis

The chemical composition was identified with the Foundry Master (WAS) optical emission spectrometer (Hitachi, Tokyo, Japan).

### 2.2. Static Tensile Tests

The static tensile tests were carried out on a Z250 testing machine from Zwick Roell (ZwickRoell GmbH & Co. KG, Ulm, Germany). The initial force was 500 N, and the tensile speed was 45 mm/min. Static tensile tests were carried out on the samples taken from the steel sheets in the rolling direction. The selection of test parameters was made in accordance with PN-EN 10002-1 [29] and AC1 (2004).

### 2.3. Drawability Tests

Drawability tests were carried out following the Erichsen IE method [30] (PN-EN ISO 20482: 2014-02) on a Tytan 20 device. The diameter of the end of drawing punch was 20 mm, while the internal diameter of the die was 27 mm. For each tested sheet, 9 measurements of the drawing depth were made.

### 2.4. Erosion Tests

The test stand, shown in Figure 1, was prepared to carry out the erosion test. Sheets with dimensions of 200 × 200 mm were used for the experiments. They were placed 180 mm from the end of the nozzle. An abrasive in the form of quartz sand with a fraction of 0.4–1.0 mm was used. An Atmos PD71 compressor (Atmoc Chrást, Chrást, Czech Republic) with a nominal pressure of 7.0 bar and a capacity of 7.6 m^3^/min was used for the tests, using air as the carrier medium. The diameter of the outlet nozzle used was 8 mm. For the first batch of tested samples, the time of the erosive impact was 60 s. Then, in the case of perforation of a given type of steel sheet, the impact time for the next test was shortened to 30 s, and when after 60 s the sample did not show a tendency to perforation, the time of the next test was extended to 120 s. The test times for individual materials are summarized in Table 1.

### 2.5. Steel Sheet Thickness at the Erosion Region

The diameters and depth of the craters at the abrasive impact region were measured using an electronic caliper (PHU GAABI, Wroclaw, Poland), according to the scheme shown in Figure 2, depending on the presence or absence of perforation. The steel sheets were weighed before and after the erosion treatment on a WPS 510 balance (Radwag, Cracow, Poland).

### 2.6. The Analysis of the Microstructure

The microstructures, in as-delivered condition, were examined using a Carl Zeiss Axiovert 200 MAT microscope (Carl Zeiss Microscopy Deutschland GmbH, Oberkochen, Germany). The average grain diameter was determined by the use of the Jeffries method. The cross-sections of the samples were etched with 2% nital. The tests were carried out on three planes (shown in Figure 5) marked as STD-LTD, LTD-RD, RD-STD and defined by the directions: RD—rolling direction, LTD—long transverse direction, STD—short transverse direction. The microstructure was also observed in the cross-section of the sheets after the erosion test in order to assess the influence of such test on the microstructure of the investigated materials.

### 2.7. Hardness of Steel Sheets

#### 2.7.1. In the Plane of Rolling before Erosion Test

Vickers hardness measurements for the investigated materials were performed under the load of 98 N (HV10). The hardness measurement time was 15 s. The tests were carried out using a TUCON 2500 Hardness Tester (Bühler AG, Esslingen am Neckar, Germany).

#### 2.7.2. After Erosion Test in the Cross-Section of the Sample and under Low Load

The test of the hardness of the steel sheets after erosion treatment for 30 and 60 s was carried out on the cross-section of the samples made of DC03 and DC05 steel. The tests were carried out using a TUCON 2500 Hardness Tester (Bühler AG, Esslingen am Neckar, Germany). The measurement points were located every 2 mm from the edge of the sample on the side of the erosive impact (Figure 3). Each of the samples was subjected to 13 measurements. The total distance of the last measurement from the erosive edge was 24 mm. The hardness was measured using the Vickers method with a load of 1.96 N (HV0.2). The measurement time was 15 s.

### 2.8. Surface Roughness Measurements

The surface of the investigated samples after the erosion test was examined using a scanning electron microscope Phenom XL (Thermo Fisher Scientific, Waltham, MA, USA).

The observations of the surface of the investigated steel sheets were made at a distance of 25 mm from the axis of the abrasive beam (in this region the extreme angle of incidence of the abrasive material was approx. 82°). The roughness of the tested steel sheets was measured at three points: 25, 50, and 75 mm from the nozzle axis, using a Veeco WykoNT9300 optical profilometer (VEECO, Los Angeles, CA, USA).

## 3. Results

### 3.1. Chemical Composition Analysis

Averaged values of the chemical compositions of the investigated steels are presented in Table 2. Comparing the measured chemical compositions for DC-type steels with the requirements of the standard showed in Table 3, it should be stated that the tested steels meet the requirements set for them; only in the case of DC05 steel, the carbon content is in the upper range provided by the standard. Similarly, in the case of DD14 steel, the measured chemical composition for this steel meets the requirements set by the standards.

### 3.2. Static Tensile Test

The results of the static tensile tests are shown in Figure 4. The basic difference in the character of the flow curves of the static tensile test of DC and DD steels was the occurrence of a clear yield point in the case of DD steel, which indicated its aged state. The time that elapsed since the production of these steels caused “decorating” of the dislocations with the so-called Cottrell atmospheres. Deep drawing such steel sheets can be very problematic. Taking the above into account, in the case of DC steels, the assumed yield point was determined, and in the case of DD steel, both the upper and lower yield points were specified.

Table 4 summarizes the obtained results of the mechanical properties obtained from the static tensile test. The yield strength obtained for DC steels is in accordance with the requirements of the standard [32]. The properties of DC05 steel were closest to those specified in the standard. However, in the case of DD steel, the lower yield point R_eL_ value is in the middle range of the values given by the standard provided for such steel. The samples made of DC04 steel are characterized by the highest yield strength, and the lowest yield strength is noticed for DC05 steel.

### 3.3. Drawability Tests

The results obtained from the Erichsen test are summarized in Table 5. It was observed that DC steels showed slightly higher drawability in relation to DD type steel.

### 3.4. The Analysis of Microstructures of the Investigated Steels before Erosion Treatment

The microstructures of the tested steels (in the 3D view) are shown in Figure 5. The analysis of the microstructures confirmed that the sheets made of DC steels were cold rolled, as evidenced by grain elongation in the direction of rolling (RD, Figure 5a–c) and lower elongation of grains in the transverse direction (LTD, Figure 5a–c). The length of the grain increases in the investigated steels in the following order: DC03 steel sheet, DC04 steel sheet, and DC05 steel sheet. In the case of the sheet made of DD steel, the grain elongation is not as clear as in the case of DC steels. This is due to the manufacturing method, i.e., hot rolling in the case of DD steel. It can be noticed that the grain size in the DD steel is much smaller as compared with that revealed in the DC steels. The results of the grain size measurements for the tested materials are summarized in Table 6.

In the case of DC steels, the analysis performed on the RD-STD and LTD-STD sections indicated grain refinement according to the increasing steel grade index. In the case of the rolling plane (RD-LTD), the results are reversed. On the basis of the above results, it can be concluded that the grains are flattened to a greater extent in the direction normal to the surface of the steel sheet, according to the following steel grade sequence: DC03, DC04, and DC05. The calculations of the grain size (mean grain cross-sectional area) for DD steel confirm that in the hot-rolled steels the grain size is significantly smaller as compared with the cold-rolled steels. In the case of DD steel, there is also a difference in the change in grain size between the longitudinal and transverse cross-sections and the rolling plane. It should be assumed that, in the case of the steel sheet made of DD14 steel, there is a significant flattening of the grains along the rolling plane, which is not clearly visible in its microstructure, but results from the grain size measurements performed. Figure 6 presents the microstructures of the investigated steels at higher magnifications. In the case of DC steels, especially in the rolling plane (RD-STD), the observations of the microstructure indicate a decreasing carbide content in the following order: DC03, DC04, and DC05 steel. It can also be observed that with an increase in the DC steel grade index, the degree of coagulation and distribution of carbides increases. In the case of DD14 steel, the degree of coagulation of carbides is significantly lower and pearlitic areas can be distinguished in the rolling plane (RD-STD).

### 3.5. The Hardness of the Steel Sheet in the Rolling Plane

The summary of the obtained results of hardness measurements is presented in Table 7.

It can be observed that DD14 steel is characterized by much higher hardness as compared with DC steels. In the case of DC steels, the hardness decreases with increasing steel grade index, but this change is within the error of measurement. One should also pay attention to the significant dispersion of hardness measurements for DD steel as compared with DD14 steel, which indicates greater homogeneity of hot-rolled steel in relation to cold-rolled steel.

### 3.6. Weight Loss Measurements

The amount of erosion wear presented in Table 8 and Figure 7 indicates an increase in weight loss along with the extension of the test duration. The diameters of the craters and their depths depending on the test time and the type of the sample are shown in Figure 8 and Figure 9. As can be seen, on the one hand, the higher hardness of DD14 steel did not result in a significant reduction in weight loss, which suggests a significant role of the microstructure change (austenitization resulting from the temperature increase caused by the erosive particles hitting the steel sheet) during the erosion impact on wear. On the other hand, the higher hardness of the DD14 steel significantly limits the depth of the crater, but not its width. This can be related to the influence of hardness on the stiffness of the material. Moreover, changes in the thickness of the material after the erosion trials were observed. The initial thickness of the sample for each alloy was 1.5 mm. After the tests, the greatest reduction in thickness was observed in the area close to the erosion impact center. The correlation of the material thickness to the distance from the erosion impact center is presented in Figure 10.

### 3.7. Surface Roughness Analysis

The surfaces of the investigated steel samples, observed with the use of a scanning electron microscope, are shown in Figure 11. The fatigue mechanism of material wear is dominant. In the case of DC03 steel, after the 30 s erosion test, a tendency for material decohesion along the grain boundaries is visible; these are also areas where abrasive particles remaining on the surface can be observed. Increasing the time of abrasive interaction to 60 s resulted in an increased share of the areas of exposed fatigue scrap (Figure 11a,b).

In the case of the DC04 steel, after the 60 s erosion test, a similar proportion of areas with exposed fatigue wear and areas showing plastic deformation of the surface layers are visible. Extending the erosion test time to 120 s caused an increase in the proportion of plastically deformed surface layer. Decohesion along the grain boundaries was also clearly visible. These surface areas are formed mostly due to the fatigue exfoliation. Thus, it can be assumed that during the erosion test, plastic deformation of the surface layer and fatigue wear occur (Figure 11c,d).

In the case of the DC05 steel, after the 30 s erosion test, the surface showing fatigue wear features dominates. Increasing the time of erosive wear to 60 s causes an increase in the area of plastically deformed surfaces, which is also related to decohesion along the grain boundaries (Figure 11e,f). For the DD14 steel, after the 60 s erosion test, the share of the surface showing fatigue wear and the plastically deformed surface are similar. In this case, compared to DC steels, the areas indicating the wear mechanism connected with abrasion are visible. Extending the time of erosive wear to 120 s, in general, does not change the proportion of the wear mechanisms intensity at the surface area resulting from fatigue and associated with plastic deformation. Abrasive wear is less visible, and the decohesion process at the grain boundaries is also visible.

The differences in the nature of wear between DC and DD steels may result from the grain size and the strain hardening of the material. On the one hand, higher strain hardening of cold-rolled steel as compared with hot-rolled steel results in the intensification of fatigue wear. On the other hand, the greater strengthening resulting from the grain refinement in DD steel as compared with DC steels results in a reduction of the fatigue mechanism and in greater influence of abrasive wear and decohesion at the grain boundaries.

The roughness assessment results for different distances from the nozzle axis (25, 50, and 75 mm) are summarized in Figure 12. By analyzing the obtained data, it can be concluded that, in the case of DC steels, regardless of the distance from the axis of erosive interaction, the extension of the erosion test time is conducive to an increase in roughness, described by the R_a_ parameter. The largest change in roughness was found for DC04 steel when the erosion time increased from 60 to 120 s. However, in the case of DD steel, the time of erosive interaction, in general, did not affect the R_a_ parameter change. For all samples, the R_a_ parameter value decreased with increasing distance from the erosive interaction center. The lowest roughness value for each selected distance from the central point of erosive interaction was found for DD14 steel. In the case of this steel, greater roughness was found for a shorter erosion test time. Therefore, it should be assumed that the longer erosive interaction caused the initiation of a new, significantly different character of wear, which was associated with the earlier formation of the fatigue cracks along the grain boundaries, and plastic deformation sealing such cracks after prolongation of the erosive interaction.

### 3.8. Hardness of Steel Sheets after Erosion Treatment in Their Cross-Section

The hardness change analysis was presented for two cold-rolled steels: DC05 and DC03. The obtained results of the hardness measurements after the erosion test are presented in Figure 13. The results indicate that the material has the highest value of hardness at the erosion impact site. By analyzing changes in hardness with the distance from the erosion axis, it can be stated that the hardness is decreasing, and from a certain point it fluctuates around a constant value. This means that erosion did not affect the hardness along the entire length of the sample, but only around the area affected by the erosion stream. The influence of extending the erosion test time on the increase in hardness is clearly visible for the range of large angles of incidence of particles (closer to the erosive stream axis). This is the area of greater absorption of energy of the erosive particles by the investigated material.

### 3.9. Qualitative Analysis of Steel Sheets Microstructure at Cross-Sections after Erosion Tests

The microstructure analysis allowed us to distinguish areas of significantly finer grain near the axis of the abrasive stream for the DC03 30-second sample (Figure 14a), for the DC03 60-second sample (Figure 14b), and for the DC05 30-second sample (Figure 14c). The occurrence of the process of dynamic recrystallization of the plastically deformed material can be noticed. Thus, the material must have been heated during the erosion test above the recrystallization temperature. At further distances from the edge of the hole created as a result of erosive penetration, more intense grain deformation (elongation) can be noticed than in the case of the initial sheet microstructure, i.e., in the DC03 30-second sample (Figure 14d), for the DC05 30-second sample (Figure 14e), and for sample DC05 60-second (Figure 14f). This indicates that the steel sheet plastic deformation took place without the dynamic recrystallization process, i.e., cold plastic deformation. Additionally, in the case of the DC03 60-second and DC05 30-second samples, the acicular nature of ferrite grains is visible on the areas affected by erosive stream, which can be associated with intense cooling in the austenitic range (Figure 14g). This indicates heating of the material as a result of erosion to the range of existence of austenite. In the case of the DC03 30-second and DC03 60-second samples, the so-called white layer can be noticed (Figure 14g), resulting from austenitization, very strong plastic deformation of austenite, and its transformation into nanocrystalline martensite without the recrystallization processes and dynamic recovery of the austenitic structure.

The above analysis indicates the occurrence of the following effects as a result of the erosive impact, considering them from the axis of the incident abrasive stream: Formation of austenite and its strong deformation;Formation of austenite and its dynamic recrystallization;Plastic deformation of ferrite and its dynamic recrystallization;Plastic deformation of ferrite and its dynamic recovery;Plastic deformation of ferrite;No influence of erosion on the microstructure in the center of the steel sheet.

Extending the time of the erosion seems to widen the zone of the austenitized material, but when the steel sheet is perforated, this effect is weakened. However, the above-mentioned effects of erosion on the material are mainly dependent on the angle of incidence of the erosive particles and the intensity of their interaction with the material.

### 3.10. Quantitative Analysis of the Microstructure

The obtained results clearly show that the influence of erosion led to a significant grain refinement at the first hardness measurement point (Figure 15). For DC03 steel, the grain size in both cases was 60 μm^2^. For the DC05 steel, the differences are more noticeable, the sample eroded for 30 s has much smaller grains than the sample eroded for 60 s. This may be due to the fact that the last sample lost most of its weight, and the pieces with fine grains were detached. Relating these results to the results of hardness measurements, the effect of grain refinement on the increase of material hardness is visible. In addition, hardness is also be influenced by strain hardening, which is very little (if any) when the recrystallization and dynamic recovery process has taken place. The share of such strengthening is certainly visible in the case of the first measurement (the smallest grains) for the DC05 60-second sample. Moreover, the DC03 material characterized by a higher carbon content obtains greater solution strengthening after accelerated cooling in the austenite range. However, it should be noted that both of these steels are characterized by such a low carbon content that the hardening of the solution has a low potential, although it is visible for the first measurement point in the DC03 60-second sample. Therefore, the main effect is grain boundary strengthening.

## 4. Discussion

The results obtained in this study allowed the assessment of microstructural changes in deep-drawing steels under the influence of intense erosive impact. With an increase in the time of the erosion stream impact, greater damage occurs, and the formed crater increases its dimensions and depth. With the identical time of the erosive test, the increase in the exponent determining the drawability of the steel can be connected with a greater susceptibility to erosion, as the material weight losses in the case of DC05 steel turned out to be greater. The hardness of the tested materials did not differ significantly from the values determined by the technical standard, and the differences of these values were within the error range of the microhardness measurements. The obtained results of the hardness measurements allow us to conclude that with an extension of the erosion impact time, the material hardness at the area of its impact increases. The DC03 steel turned out to be harder both after erosion and in as-received condition. At the same erosion test time of 30 s, the DC03 and DC05 steels showed similar hardness. The longer erosion test time led to a much greater strengthening of the DC03 steel as compared with the DC05 steel (however, it may be due to the larger penetration hole in the DC05 steel). This means that the material hardens along with an extension in the erosion time. The hardness of each sample oscillates around a constant value, which means that the effects of erosion are visible only in the vicinity of the area influenced by the erosion stream. It can be assumed that the increase in hardness also results from much larger plastic deformations during the 60 s test.

The analysis of the microstructure of the investigated materials clearly shows that there is a difference in the grain size for the steel sheet subjected to the erosion treatment. Each of the samples underwent significant grain refinement at the erosion axis. This refinement became greatest at the edges from the side of the erosive stream. In addition, high temperature (above the recrystallization temperature), caused by friction and impact, and rapid cooling of the samples after the end of the test, resulted in the formation of non-equilibrium structures. For DC03 steel, regardless of the erosive test duration, the grains at the crater are very fine, similar in size, and significantly different from the grains in the rest of the material volume. At a short distance from the erosion axis, elongated, deformed grains resulting from strong cold plastic deformation are visible in the areas that have been heated below the recrystallization temperature. The DC05 steel tested for 30 s has much smaller grains than in the rest of the steel sheet, but they are larger than in the case of the DC03 steel. The DC05 60-second sample had the largest grain; however, a large weight loss signifies that the fragments with fine grains have detached. The remaining parts of the samples are characterized by large ferrite grains with clearly visible grain boundaries, which confirms that they did not undergo any changes due to the increased distance from the erosion stream. These results indicate that the erosion causes a significant grain refinement, especially in the areas where the abrasive particles directly hit. In addition, large amounts of heat are released (sufficient to heat the steel to the range of austenite existence) during erosion, leading to phase changes. Plastic deformation of the sheet leads to decohesion of the material and elongation and deformation of the material grains. The above phenomena allow the conclusion that at the edges of the erosion crater, a strongly deformed austenite is formed, which is gradually replaced by plastically deformed ferrite. The ferrite plastically deformed at a distance closer to the erosion axis dynamically recrystallizes; however, going further from the erosion stream, dynamic recovery and plastic deformation occur. In zones not affected by the erosive stream, no signs of the influence of erosion on the microstructure of the material can be seen.

By comparing the hardness test results to the microstructure allowed us to determine the relationship between the grain size and shape, and hardness. The hardness increased with an increase in microstructure refinement. It can be noticed especially at the edges of the samples at the areas influenced by erosion, where the grains are smallest, and the hardening related to the abrasive impacts is the greatest. It can be related to the dynamic recrystallization and recovery occurring in the material. In addition, the phenomena causing the material strengthening are associated with the phase change, resulting in austenitized material and strain hardening.

## 5. Conclusions

On the basis of the conducted experiments, the following conclusions can be drawn:(1)An increase in strength properties increases the resistance to erosive wear.(2)Significant results of the erosive treatment are microstructural changes affecting the material properties.(3)There is a correlation between the microstructure and the erosive wear mechanisms.(4)The erosive wear mechanisms and their intensity obviously change with the distance from the erosive stream axis.(5)There is a correlation between an increase in hardness and an increase in surface roughness with the grain refinement, and with a reduction in the sheet thickness (mass wear).(6)Extending the time of erosive impact does not change both the nature of changes in the investigated material and aerological parameters; the distance from the erosive stream axis only changes their intensity.

## Figures and Tables

**Figure 1 materials-14-03540-f001:**
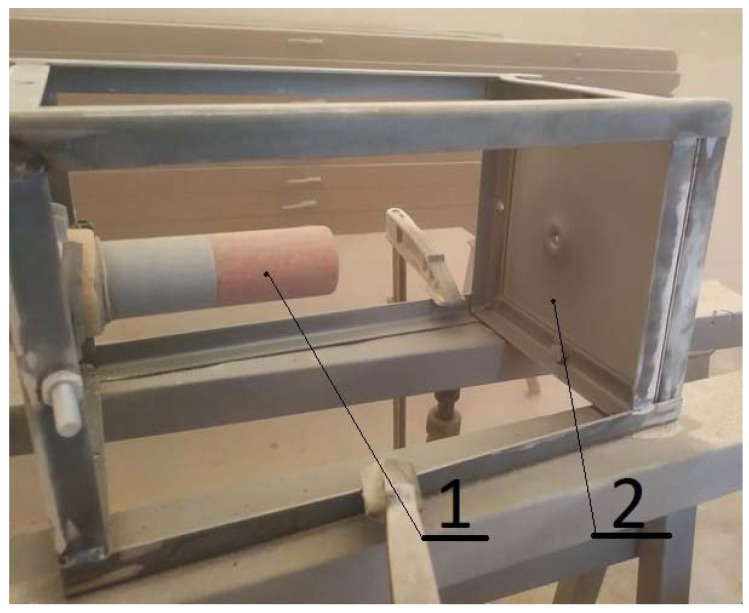
Test stand for erosion tests: (**1**) Nozzle; (**2**) tested sheet.

**Figure 2 materials-14-03540-f002:**
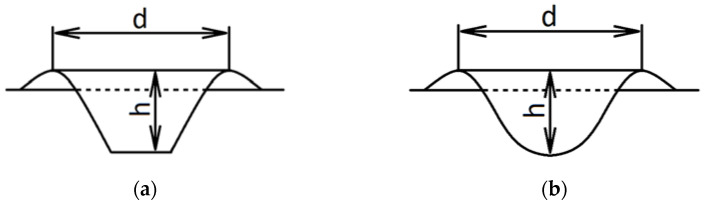
The method of performing deformation measurements after the steel sheet erosion test: (**a**) With perforation; (**b**) without perforation. d—diameter and h—height.

**Figure 3 materials-14-03540-f003:**
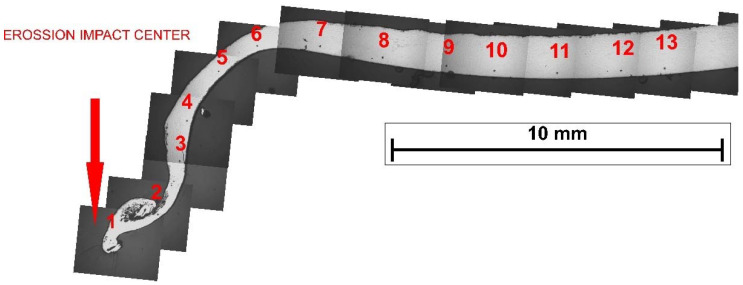
The DC05 sample after the 30 s erosion test subjected to hardness measurements: (**1**–**13**) Measuring points.

**Figure 4 materials-14-03540-f004:**
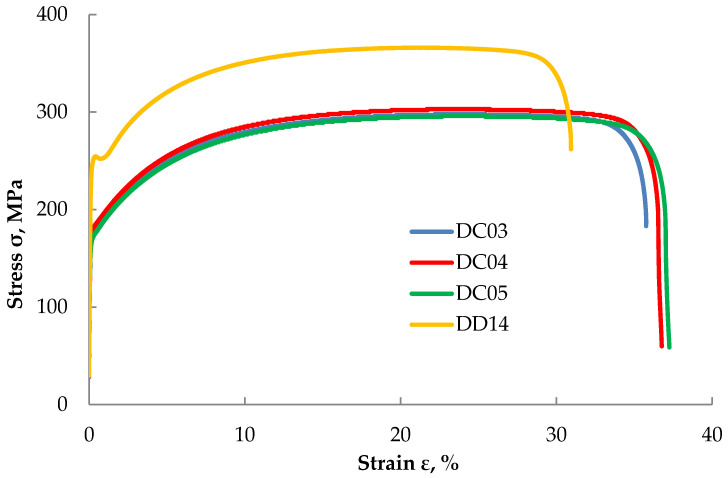
Tensile stress-strain relationship for the investigated steel sheets.

**Figure 5 materials-14-03540-f005:**
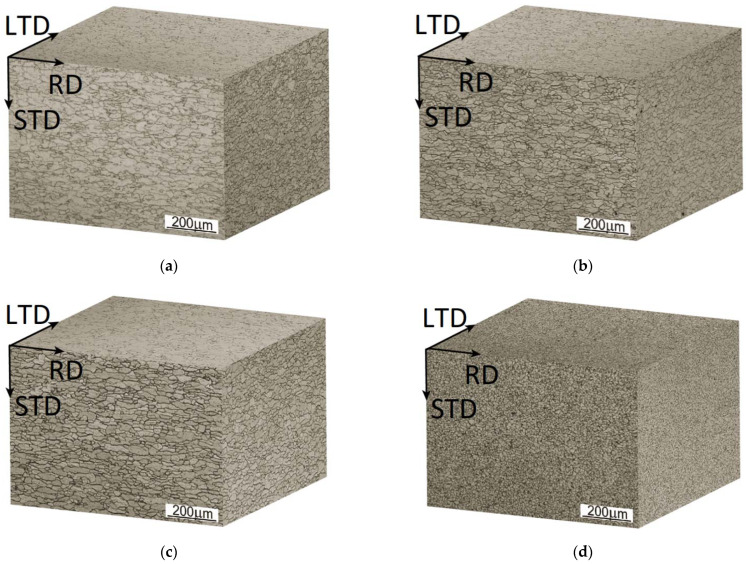
3D images of the microstructure of the investigated steel sheets: (**a**) DC03; (**b**) DC04; (**c**) DC05; (**d**) DD14. RD—rolling direction; LTD—long transverse direction; STD—short transverse direction.

**Figure 6 materials-14-03540-f006:**
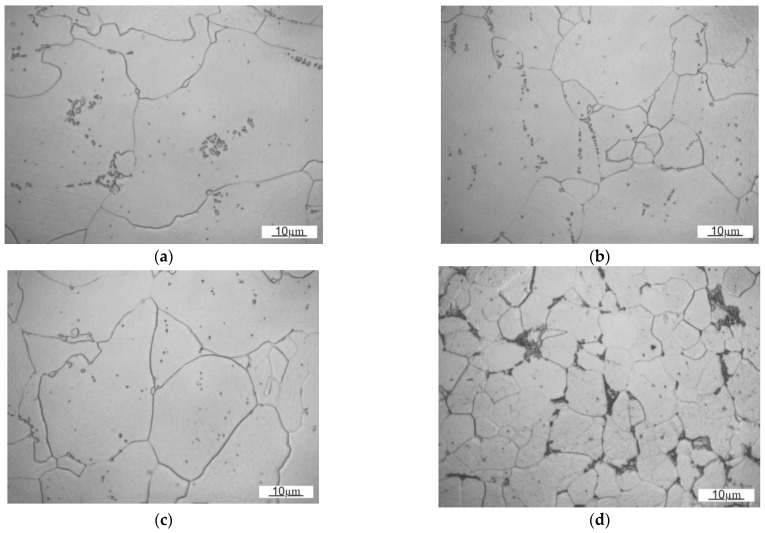
The microstructures of the investigated steel sheets in the rolling plane (RD-STD): (**a**) DC03; (**b**) DC04; (**c**) DC05; (**d**) DD14.

**Figure 7 materials-14-03540-f007:**
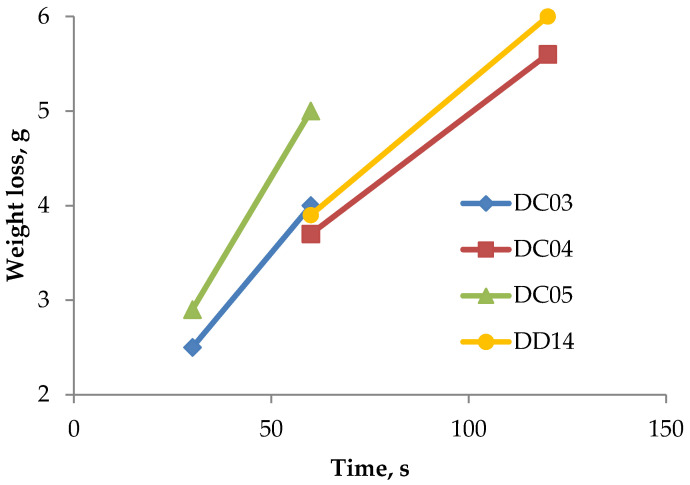
Weight loss of the investigated samples after appropriate test durations.

**Figure 8 materials-14-03540-f008:**
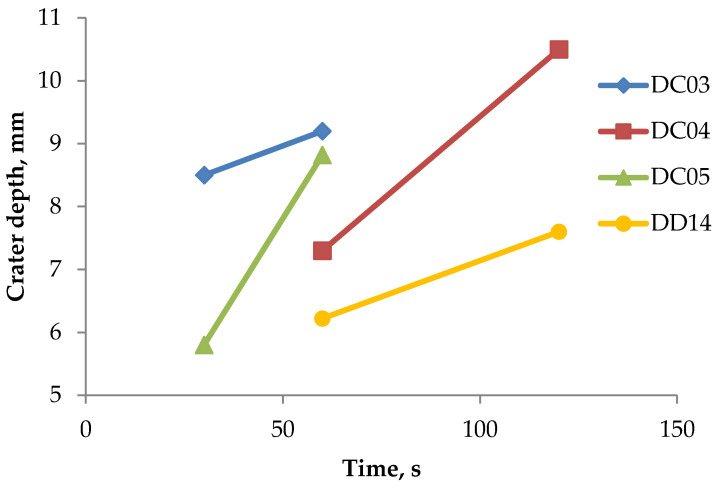
Crater depths in the investigated samples.

**Figure 9 materials-14-03540-f009:**
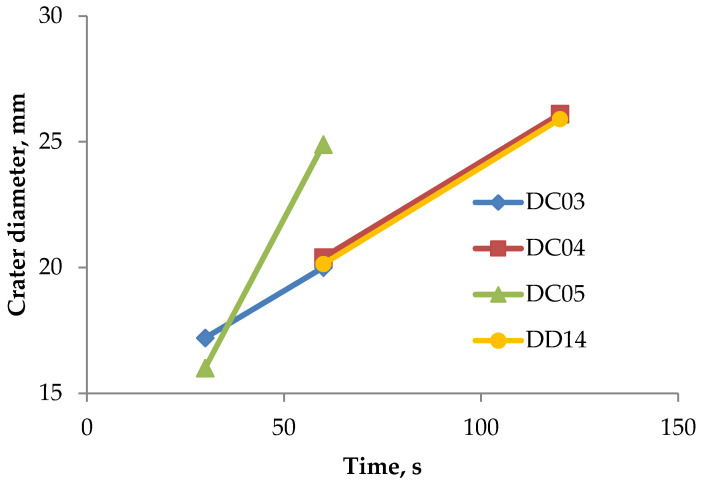
Crater diameter in the investigated samples.

**Figure 10 materials-14-03540-f010:**
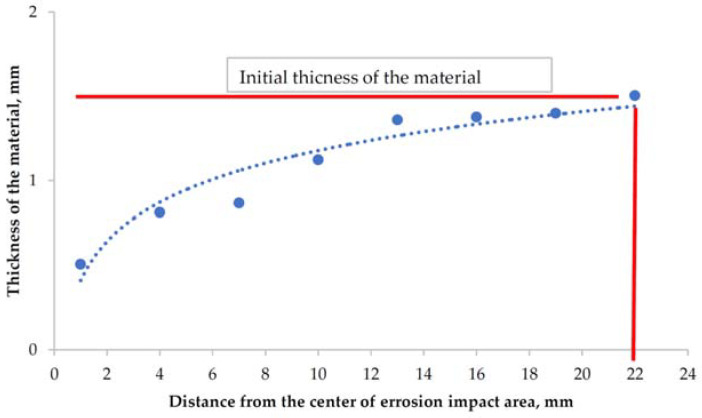
Changes in the average material thickness of the investigated samples.

**Figure 11 materials-14-03540-f011:**
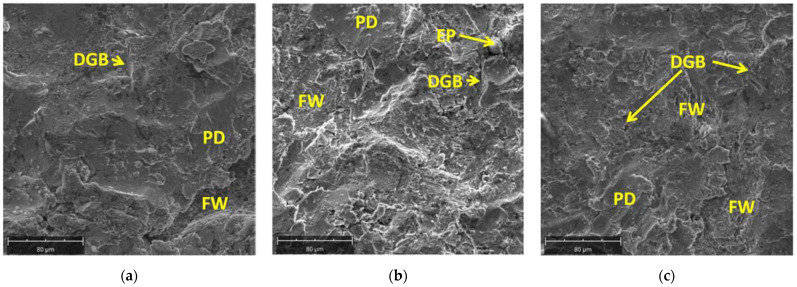
The surfaces of the investigated steel samples at a distance of 25 mm from the crater axis: (**a**) DC03, 30-second erosion test; (**b**) DC03 60-second erosion test; (**c**) DC04 60-second erosion test; (**d**) DC04 120-second erosion test; (**e**) DC05 30-second erosion test; (**f**) DC05 60-second erosion test; (**g**) DD14 60-second erosion test; (**h**) DD14 120-second erosion test. FW—fatigue wear; PD—plastic deformation; DGB—decohesion along the grain boundaries; EP—erosive particle stuck into the surface of the sheet.

**Figure 12 materials-14-03540-f012:**
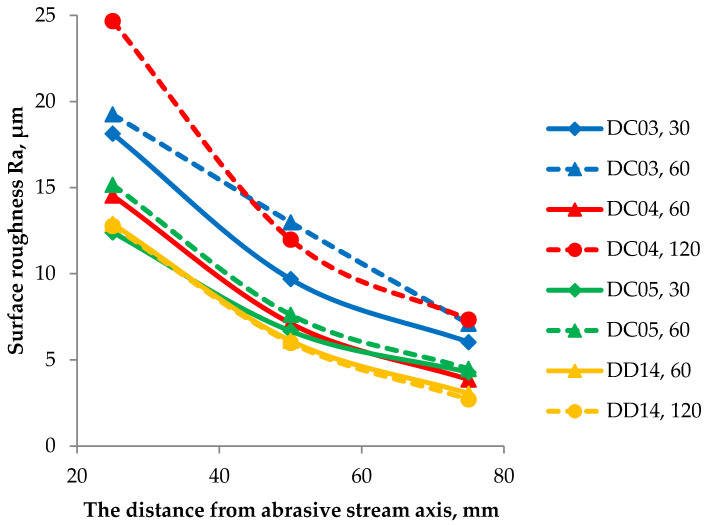
Surface roughness R_a_ in relation to the distance of the sample from the crater axis for erosion test times of: 30 s, 60 s and 120 s.

**Figure 13 materials-14-03540-f013:**
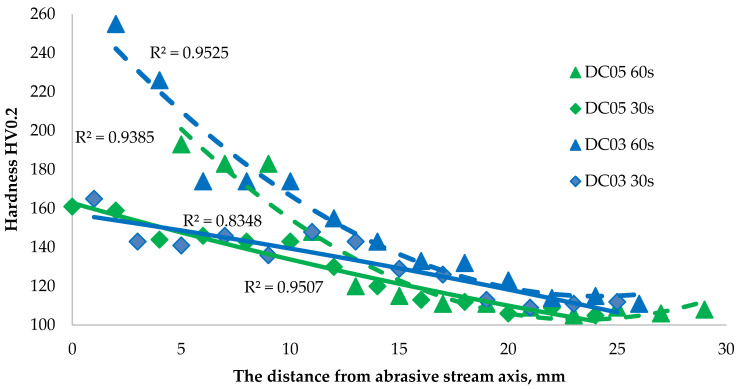
Hardness of the investigated steel cross-section in relation to the distance of the sample from the axis of the abrasive stream.

**Figure 14 materials-14-03540-f014:**
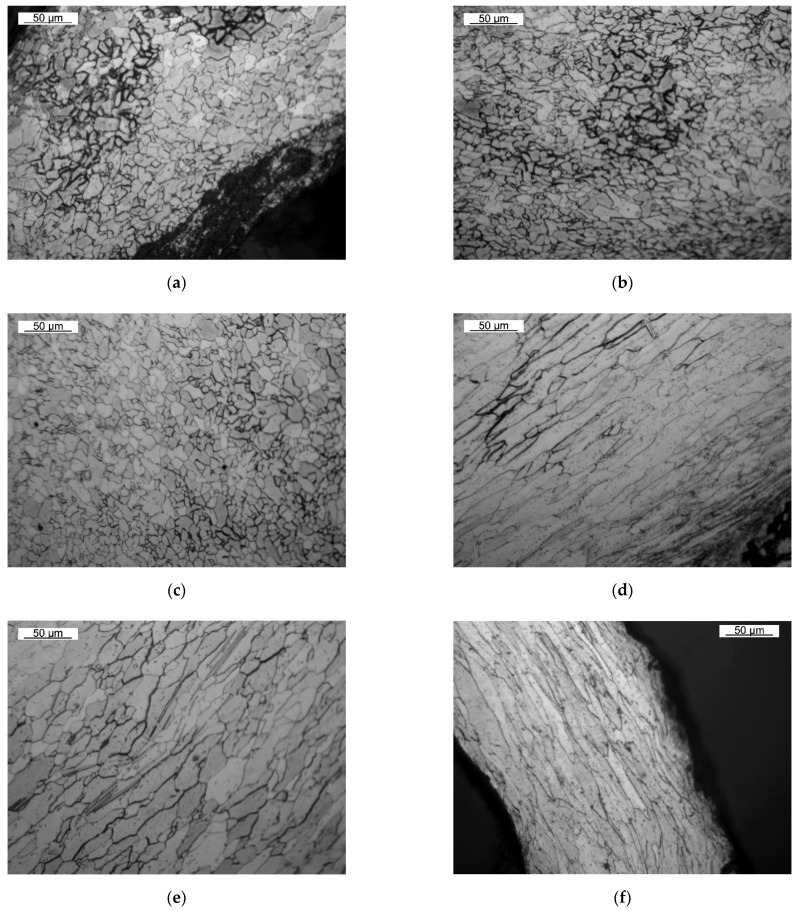
The microstructure of the investigated samples at the area of erosion wear: (**a**,**d**,**h**) DC03 30-second sample; (**b**,**g**) DC03 60-second sample; (**c**,**e**) DC05 30-second sample; (**f**) DC05 60-second sample.

**Figure 15 materials-14-03540-f015:**
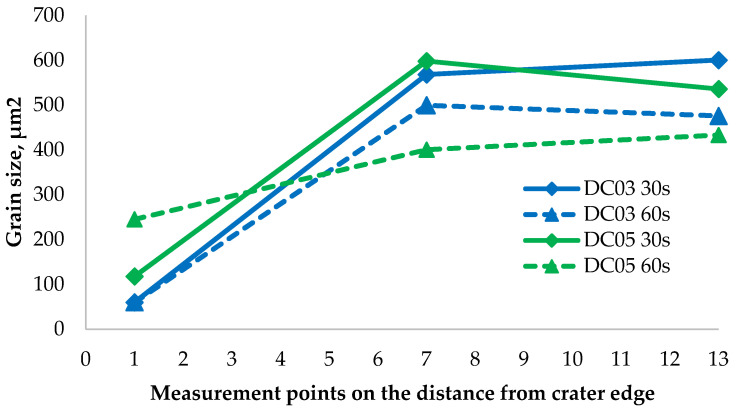
The relationship of the average grain size for measurement points: 1 (2 mm from crater edge), 7 (14 mm from crater edge), 13 (26 mm from crater edge).

**Table 1 materials-14-03540-t001:** Time of the erosion tests performed for individual steel grades.

	Erosion Time, s		
Steel Grade	30	60	120
DC03	×	×	
DC04		×	×
DC05	×	×	
DD14		×	×

**Table 2 materials-14-03540-t002:** The chemical composition of the investigated steels (optical emission spectroscopy).

	Chemical Composition, wt. %							
	C	Mn	P	S	Si	Cr	Al	Cu
DC03	0.070 ± 0.019	0.21 ± 0.01	0.015 ± 0.003	0.012	<0.005	0.02	0.057	0.02
DC04	0.071 ± 0.026	0.24 ± 0.02	0.012 ± 0.001	0.007	0.06	0.02	0.076	0.04
DC05	0.061 ± 0.016	0.22 ± 0.01	0.011 ± 0.001	0.005	0.07	0.01	0.067	0.02
DD14	0.078 ± 0.007	0.30 ± 0.01	0.011 ± 0.002	0.008	<0.005	0.01	0.054	0.03

**Table 3 materials-14-03540-t003:** The standardized maximum content of alloying elements in the investigated steel grades, wt. % [31].

	Chemical Element, wt. %			
Type of Steel	C	P	S	Mn
DC03	0.10	0.035	0.035	0.45
DC04	0.08	0.030	0.030	0.40
DC05	0.06	0.025	0.025	0.35
DD14	0.08	0.025	0.025	0.35

**Table 4 materials-14-03540-t004:** The mechanical properties of the investigated steels determined in tensile tests.

	DC03	DC04	DC05	DD14
R_p0.2_, MPa	175	180	172	-
R_eH_, MPa	-	-	-	251
R_eL_, MPa	-	-	-	249
R_m_, MPa	298	303	296	361
E, GPa	197	207	193	205

R_p0.2_—yield point; R_eH_—upper yield point; R_eL_ – lower yield point; R_m_—tensile strength; E—Young’s modulus.

**Table 5 materials-14-03540-t005:** The results for the measurements of the depth of the draw.

	DC03	DC04	DC05	DD14
Depth of the draw, mm	12.85 $\begin{array}{c} +0.11\\ -0.13 \end{array}$	12.65 $\begin{array}{c} +0.05\\ -0.09 \end{array}$	12.86 $\begin{array}{c} +0.04\\ -0.05 \end{array}$	12.57 $\begin{array}{c} +0.11\\ -0.07 \end{array}$

**Table 6 materials-14-03540-t006:** The size of the grain cross-sectional area in the tested steels.

	Plane RD-STD, µm^2^	Plane LTD-STD, µm^2^	Plane RD-LTD, µm^2^
DC03	399 ± 20	432 ± 22	411 ± 21
DC04	370 ± 18	393 ± 20	648 ± 32
DC05	295 ± 15	360 ± 18	786 ± 39
DD14	66 ± 3	68 ± 3	115 ± 6

**Table 7 materials-14-03540-t007:** Hardness HV10 of the investigated steel sheets.

DC03	DC04	DC05	DD14
77 $\begin{array}{c} +12\\ -17 \end{array}$	72 $\begin{array}{c} +10\\ -10 \end{array}$	70 $\begin{array}{c} +7\\ -8 \end{array}$	106 $\begin{array}{c} +1\\ -2 \end{array}$

**Table 8 materials-14-03540-t008:** Crater diameter (according to Figure 2) (mm)/crater depth and weight loss in (g).

Steel	30 s		60 s		120 s	
	mm	g	mm	g	mm	g
DC03	17.2/8.5 *	2.5	20/9.2 *	4.0	-	-
DC04	-	-	20.4/7.3 *	3.7	26.1/10.5 *	5.6
DC05	16/5.8 *	2.9	24.89/8.82 *	5.0	-	-
DD14	-	-	20.14/6.22	3.9	25.9/7.6	6.0

*—perforation.

## Data Availability

The data presented in this study are available on request from the corresponding author.

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
