# Peer review of "The Effect of Sandblasting on Properties and Structures of the DC03/1.0347, DC04/1.0338, DC05/1.0312, and DD14/1.0389 Steels for Deep Drawing"

_materials, 2021, doi:10.3390/ma14133540_

Round 1
Reviewer 1 Report
This manuscript studied the influence of sand blasting on properties and structures of various types deep drawing steels. Material tests and microstructure analysis were applied. Some comments should be addressed after reviewing this study:
1 As the introduction said: “but the information concerning the influence of the material surface behaviour resulting from sandblasting on its properties and wear behawior (I think this should be behaviour?) is still limited.” However, many studies focused on the surface integrity of the material after sand blasting. Thus the research gap, or the novelty of this work, must be further and clearly explained.
2 In section 2.2, any references on the choice of the testing boundary parameters? In figure 1: Test stand for erosion tests, the reviewer believes that annotations could be added to explain or highlight for readers to easily understand, same as in figure 10, 11 and 14.
3 Line 259: “The fatigue mechanism of material wear is dominant” seems a little unclear to me. Please clarify.
4 In the discussion, the authors presented the phenomenon of the tests, the reviewer believes that some suggestions could be raised on the material strengthening, anti-fatigue design, process optimisations, etc.
Author Response
Dear Reviewer,
Thank you very much for taking the time to read our manuscript thoroughly and make recommendations for its correction and improvement. We have read the comments carefully and have responded to all your comments.
Remark 1
Does the introduction provide sufficient background and include all relevant references?
Response: Thank you for this notice. The introduction was enriched by additional analysis of the research focused on the influence of sandblasting on the development of the properties of steels. Appropriate references were introduced into the manuscript.
Remark 2
As the introduction said: “but the information concerning the influence of the material surface behaviour resulting from sandblasting on its properties and wear behawior (I think this should be behaviour?) is still limited.” However, many studies focused on the surface integrity of the material after sand blasting. Thus the research gap, or the novelty of this work, must be further and clearly explained.
Response: Thank you for this notice. This research was mainly focused on the analysis of the influence of sandblasting on the development of the resulting properties and wear behaviour of deep drawing steels and looking through all accessible sources of information we noticed that there is a lack of information concerning this issue. However, the introduction was enhanced by additional information on the research focused on the influence of sandblasting on the development of the properties of steels and appropriate references were incorporated into the manuscript.
Remark 3
In section 2.2, any references on the choice of the testing boundary parameters?
Response: The selection of test parameters was made in accordance with PN-EN 10002-1 and AC1 (2004) – it was added to the text.
Remark 4
In figure 1: Test stand for erosion tests, the reviewer believes that annotations could be added to explain or highlight for readers to easily understand,
Response: It was added to the Figure 1
Remark 5
same as in figure 10, 11
Response: It was added to the figure 11. Figure 10 was deleted.
Remark 6
Line 259: “The fatigue mechanism of material wear is dominant” seems a little unclear to me. Please clarify.
Response: It is based on multiple deformation of the surface layer by the abrasive particles. This causes cracks and peeling of the material.
Remark 7
In the discussion, the authors presented the phenomenon of the tests, the reviewer believes that some suggestions could be raised on the material strengthening, anti-fatigue design, process optimisations, etc.
Response: The way of improving the properties of the investigated materials could be hardening by solutioning in the range that does not reduce the impact toughness and drawability, e.g. by adding Ni.
Reviewer 2 Report
This paper investigates the effect of Sand Blasting on Properties and Structures of four different type of steels, which provides additional knowledge on understanding the evolution of structures and associated properties during sand blasting. The experimental are well done and the paper is in good shape. However, some conclusions and statements are not clearly supported by the evidence and arguments presented in the paper. Some more comments are given below.
In figure 3, the image indicates significant deformation of the sample after erosion. The thickness of the sample become non-uniform along the sample. What is the initial thickness of the samples? Please indicate the erosive point in the image.
In Table 8, please indicate if there is perforation for specific samples.
In line 242 and throughout the manuscript, the author states that phase change occurred, i.e. austenitization resulting from the temperature increase caused by the erosive particles hitting the steel sheet. However, there is no solid evidence to support this statement. Does the author have EBSD or XRD results to further prove this statement?
In section 3.7, the author claimed that different deformation mechanisms occurred after erosion, including fatigue wear, plastic deformation, decohesion along the grain boundaries. However, it is hard to see the difference from Fig. 10 and Fig. 11. Please give more details and mark the different areas in the images.
In Fig. 14, the caption is not in English. Please also specify in which position you take these images.
In line 367, The above analysis indicates the occurrence of the following effects. Little evidences in given to support these effects.
The conclusion part is missing.
Author Response
Dear Reviewer,
Thank you very much for taking the time to read our manuscript thoroughly and make recommendations for its correction and improvement. We have read the comments carefully and have responded to all your comments.
Remark 1
In figure 3, the image indicates significant deformation of the sample after erosion. The thickness of the sample become non-uniform along the sample. What is the initial thickness of the samples? Please indicate the erosive point in the image.
Response: The drawing has been made and put in the place of the old Figure 10
Remark 2
In Table 8, please indicate if there is perforation for specific samples.
Response: The information has been added
Remark 3
In line 242 and throughout the manuscript, the author states that phase change occurred, i.e. austenitization resulting from the temperature increase caused by the erosive particles hitting the steel sheet. However, there is no solid evidence to support this statement. Does the author have EBSD or XRD results to further prove this statement?
Response: EBSD and XRD studies were not performed. The statement was based on the acicular ferrite morphology, which indicates a given phenomenon (intense cooling in the austenite range). Due to the low content of carbon, XRD tests are unlikely to give an unequivocal result. Perhaps the EBSD study would explain the transformation phenomenon, but a similar result is to be expected for the recrystallization process. However, this research would be very time-consuming without being sure of getting an answer. The solution could be to measure the temperature during the erosion test and relate it to the characteristic temperatures of the tested steels. However, this requires the design of a new test methodology. This will be performed in future research.
Remark 4
In section 3.7, the author claimed that different deformation mechanisms occurred after erosion, including fatigue wear, plastic deformation, decohesion along the grain boundaries. However, it is hard to see the difference from Fig. 10 and Fig. 11. Please give more details and mark the different areas in the images.
Remark 5
In Fig. 14, the caption is not in English.
Response: The caption has been translated
Remark 6
In line 367, The above analysis indicates the occurrence of the following effects. Little evidences in given to support these effects.
Remark 7
The conclusion part is missing.
Response: The conclusion has been added
Reviewer 3 Report
Manuscript Number: Manuscript ID materials-1246496 entitled:
The Effect of Sand Blasting on Properties and Structures of the DC03/1.0347, DC04/1.0338, DC05/1.0312 and DD14/1.0389 Steels for Deep Drawing
General comment
The paper presents the results of the analysis of the influence of abrasive impact erosion on surface and properties of DC03/1.0347, DC04/1.0338, DC05/1.0312 and DD14/1.0389 deep drawing steels and the obtained results indicate that the erosive impact may cause a significant grain refinement of the microstructure and material phase changes. This research expands the knowledge on specific mechanisms taking place during sand blasting and their influence on deep drawing steels properties and wear behavior.
Some recommendations and observation are listed below:
- Regarding Materials and Methods the authors write:
“The steels used in the experiments were in an aged condition, that is, more than 6 months elapsed from the date of production to the performance of the tests. The material in this condition was selected due to the purpose of determining the possibility of the regeneration (repair) process of structural elements made of such steels.”
For a better understanding, it would be good to detail briefly the storage conditions (for these 6 months from the date of production), which led to an aging condition of the material. If the steel was apriori used for other purpose or just stored more or less properly?
- In Figure 7. The weight loss of the investigated samples after appropriate test durations, were presented, but some values are not in accordance with Table 8 (DC05 at 60 s is about 5 and not 5.5).
- Figures 7, 8 and 9 may be better inserted in a section, additionally, because they are presented in the specified tables. Erosion rates of the steel specimens can be determined from the slope of the weight loss versus time in (mm/year) and are more suggestive and can be discussed based on the properties presented in Table 4.
- The roughness of the eroded surface is related with the mass detachment process under erosion conditions. An approach of the roughness profile in correlation to the volume removed can give more suggestive results (for example, the volume loss in each case divided by the maximum volume loss for a unit-based normalization to the maximum value).
- Section 3 is entitled Results and discussion and section 4 is also entitled Discussion.
Correct the section 3 only with Results.
- Explain briefly the reason to present hardness change analysis only for these two cold rolled steel samples: DC05 and DCO3 (in paragraph 3.8)?
- Compare the results in the current work with other / or similar erosion results reported in literature.
- Conclusions: This section is not mandatory, but can be added to the manuscript if the discussion is unusually long or complex.
In my opinion, some conclusions (or main conclusion) are necessary and welcome to highlight the purpose of this study.
- In References section some points and comma are missing or doubled (ref 4, ref 13…)
Author Response
Dear Reviewer,
Thank you very much for reviewing our manuscript. According to the comments and the questions, we have carefully revised the article text. Below we answer to all the remarks. The changes in the manuscript have been highlighted
Remark 1
Regarding Materials and Methods the authors write: “The steels used in the experiments were in an aged condition, that is, more than 6 months elapsed from the date of production to the performance of the tests. The material in this condition was selected due to the purpose of determining the possibility of the regeneration (repair) process of structural elements made of such steels.” For a better understanding, it would be good to detail briefly the storage conditions (for these 6 months from the date of production), which led to an aging condition of the material. If the steel was apriori used for other purpose or just stored more or less properly?
Remark 2
In Figure 7. The weight loss of the investigated samples after appropriate test durations, were presented, but some values are not in accordance with Table 8 (DC05 at 60 s is about 5 and not 5.5).
Remark 3
Figures 7, 8 and 9 may be better inserted in a section, additionally, because they are presented in the specified tables. Erosion rates of the steel specimens can be determined from the slope of the weight loss versus time in (mm/year) and are more suggestive and can be discussed based on the properties presented in Table 4.
Response: The drawing has been made and put in the place of the old Figure 10 for better understanding the results
Remark 4
The roughness of the eroded surface is related with the mass detachment process under erosion conditions. An approach of the roughness profile in correlation to the volume removed can give more suggestive results (for example, the volume loss in each case divided by the maximum volume loss for a unit-based normalization to the maximum value).
Response: Thank you for your attention. However, the reference of roughness to the mass in this case does not make sense, because the roughness of the surface depends on a specific place, and erosion is not homogeneous
Remark 5
Section 3 is entitled Results and discussion and section 4 is also entitled Discussion. Correct the section 3 only with Results.
Response: It was corrected in the article
Remark 6
Explain briefly the reason to present hardness change analysis only for these two cold rolled steel samples: DC05 and DCO3 (in paragraph 3.8)?
Response: A detailed analysis of the correlation of changes in microstructure and hardness required focusing on the most important aspects. The most different steels were chosen in terms of the chemical composition, but produced in the same way and subjected to the same procedure of the erosion tests.
Remark 7
Compare the results in the current work with other / or similar erosion results reported in literature.
Response: Such a comparison is difficult due to the non-standard material for such interactions. It can only be concluded that qualitatively the mechanisms of erosive wear in relation to the erosive particle incidence angle are similar to those presented in the literature (e.g. in relation to the fatigue wear). Here, however, we suggest a phase change and a bending of the sheet surface occurring near its perforation, which changes the angle of the particles interaction.
Remark 8
Conclusions: This section is not mandatory, but can be added to the manuscript if the discussion is unusually long or complex.
Response: The conclusion has been added to the article
Remark 9
In References section some points and comma are missing or doubled (ref 4, ref 13…)
Response: Thank you for this notice. The references were revised and corrected.
Round 2
Reviewer 1 Report
Comments has been addressed, which are acceptable.
Reviewer 2 Report
Authors convincingly enough replied to my questions and properly improved the manuscript. Therefore, I am glad to recommend the revised manuscript for publication.